# Experimental Validation of High Spatial Resolution of Two-Color Optical Fiber Pyrometer

**DOI:** 10.3390/s23094320

**Published:** 2023-04-27

**Authors:** Sahar Safarloo, Alberto Tapetado, Carmen Vázquez

**Affiliations:** Electronics Technology Department, School of Engineering, Carlos III University of Madrid, 28911 Leganés, Spain; ssafarlo@ing.uc3m.es (S.S.); atapetad@ing.uc3m.es (A.T.)

**Keywords:** multi-mode fiber, optical fiber, single mode fiber, spatial resolution, temperature measurement, two-color optical fiber pyrometer

## Abstract

Taking non-contact temperature measurements in narrow areas or confined spaces of non-uniform surfaces requires high spatial resolution and independence of emissivity uncertainties that conventional cameras can hardly provide. Two-color optical fiber (OF) pyrometers based on standard single-mode (SMF) and multi-mode optical fibers (MMF) with a small core diameter and low numerical aperture in combination with associated commercially available components can provide a spatial resolution in the micrometer range, independent of the material’s emissivity. Our experiment involved using a patterned microheater to generate temperatures of approximately 340 °C on objects with a diameter of 0.25 mm. We measured these temperatures using two-color optical fiber pyrometers at a 1 kHz sampling rate, which were linearized in the range of 250 to 500 °C. We compared the results with those obtained using an industrial infrared camera. The tests show the potential of our technique for quickly measuring temperature gradients in small areas, independent of emissivity, such as in microthermography. We also report simulations and experiments, showing that the optical power gathered via each channel of the SMF and MMF pyrometers from hot objects of 250 µm is independent of distance until the OF light spot becomes larger than the diameter of the object at 0.9 mm and 0.4 mm, respectively.

## 1. Introduction

Temperature measurements are crucial for studies of heat generation and transfer processes in a variety of engineering systems. Many of these systems, including microscale engineering systems, have dimensions of only a few micrometers or even tens of nanometers [1]. It has been shown that temperature measurements in earthquake physics [2], materials processes [3], diamond cutting [4], and hypersonic vehicles [5] also need to have a spatial resolution in the order of micrometers. Without a high spatial resolution and fast temperature measurement techniques, it is impossible to conduct experimental studies of the microscale thermal processes that occur in these systems.

Generally, temperature measurement technologies can be grouped into contact measurements, including thermocouples and resistance thermometers, and non-contact techniques, which include infrared (IR) cameras, pyrometers, and micro-Raman thermometers [6].

In the case of contact sensors, placing them in hard-to-reach measuring areas is challenging [7]. Moreover, thermocouple temperature measurements are significantly influenced by the ambient temperature and the length of the sensor [7,8], in addition to the fact that they only measure average temperatures across the entire length of the sensor and do not capture localized maximum temperatures [9].

The IR camera is a non-contact technology that has been applied to a wide range of applications, such as detecting the temperature of the side of a tool’s face during metallurgical processes [10], metal additive manufacturing [11], and nuclear fusion [12]. However, as mentioned in [13], emissivity uncertainty has a negative impact on its accuracy. The minimum measurable spot size on IR cameras is limited by its field of view and depends on the distance, reaching roughly 1 mm. This feature can be enhanced to 5–10 µm using an additional close-up lens at a higher cost [13,14]. As mentioned, an important disadvantage of the IR camera is that measurement uncertainty depends on the object’s emissivity, which can be lightly compensated at certain temperature ranges by performing regular calibrations for each kind of material and measuring condition [15].

On the other hand, micro-Raman thermography is commonly utilized in microelectronics to measure device temperatures. Since a laser beam is used to excite the measuring zone in micro-Raman thermography, the spatial resolution is dependent on the laser beam spot and is typically in the sub-micrometer range [16]. However, the acquisition time for this measuring technique is long and might vary depending on the material being studied from a few seconds to several minutes [17].

Compared to the methods mentioned above for temperature sensing, the two-color optical fiber pyrometer not only has the advantages of a fast response [18], highly localized temperature measurement [19], and measurement accuracy [20], but it can also be used in wet conditions [6] and provides fast and localized measurements, for example in the machining of metals [21]. This technique can integrate a self-reference method by using two wavelength bands, such as in [22], allowing an absolute temperature measurement to be obtained without knowing the material’s emissivity [23]. It is achieved by choosing bands that are far enough apart to enable filtering and lower measurement errors, yet close enough together to avoid the emissivity effect of the material [18,19]. The spatial resolution of the pyrometer is dependent on the optical fiber employed and the distance between the optical fiber and the object being measured and is limited by the diffraction limit of light. In the literature, various optical fibers have been proposed. These pyrometers typically employ optical fibers with core sizes ranging from 280 µm [24] to 1 mm [23]. The utilization of adjacent channel bands as suggested in [18,19] facilitated the deployment of indistinguishable photodetectors across both channels. The aforementioned approach, in combination with a demultiplexing technique that incurs minimal insertion loss, enabled the utilization of optical fibers possessing core diameters as small as 62.5 µm. A standard single-mode fiber was utilized in [20] for the first time and could theoretically achieve a spatial resolution as small as 16 µm for a target surface at 25 µm [25]. Table 1 provides information on the estimated diameter of the smallest spot size that can be measured using different pyrometers. However, it is important to note that none of the authors of the studies mentioned above have conducted any actual measurements to verify the spatial resolution capabilities of these systems.

In this work, the authors have tested, for the first time, the high spatial resolution of optical fiber pyrometers using single-mode and multi-mode standard fibers with a patterned heated sample using the same conditioning and acquisition system as that in [20] and compared them with the results of a commercial IR camera. We have also proved the relationship between distance and the spatial resolution of the pyrometer via experiments and simulations.

## 2. Theoretical Background and Modeling

All physical objects emit electromagnetic radiation in every direction at a certain temperature (T). The quantity of this radiation, called spectral radiance (L), is the basis of pyrometry. Planck’s law can be used to estimate the radiance that is emitted by objects, as follows:(1)$$L\left(\lambda ,T\right)=\frac{C_{1}\times \varepsilon \left(\lambda ,T\right)}{\lambda^{5}\times \left(e^{\frac{C_{2}}{\lambda \cdot T}}-1\right)}$$
where ε is the emissivity of the measured object, and C_1_ and C_2_ are Planck’s radiation constants, with values of 1.19 × 10^8^ W·Sr^−1^·μm^4^·m^−2^ and 1.44 × 10^4^ μm·K, respectively.

The target’s radiance captured with a two-color optical fiber pyrometer is separated into two wavelength bands. The acceptance cone of the optical fiber, which is determined by the numerical aperture (NA) of the fiber and the distance from the target surface, serves as the limit for the spectral radiance measured in optical fiber pyrometers. The NA depends on the refractive indices of fiber cladding and core. For a circular target whose center is aligned with the axis of the optical fiber, see Figure 1, and whose placement is perpendicular to the optical fiber, the measured current signal, I_D_, of the photodetector for each wavelength band is given by [18]:(2)$$I_{D}\left(T\right)=\int_{\lambda =\lambda_{A}}^{\lambda_{B}}R\left(\lambda \right)\times \mathrm{IL}\left(\lambda \right)\times \alpha \left(\lambda \right)\cdot \int_{S_{T}}^{}\int_{A}^{}L\left(\lambda ,T\right)\times \mathrm{dA}\times {\mathrm{dS}}_{o}\times d\lambda$$
where λ_A_ and λ_B_ are the shortest and longest wavelengths of each wavelength band, respectively, R is the photodetector responsivity, IL is the insertion loss of the filter, α is the fiber attenuation coefficient, and dA and dS_o_ are the differential elements of the solid angle and the target surface, respectively.

By integrating the radiation emitted by each dS_o_ of an object of radius r_T_ with the cone surface projected by the fiber NA, the power acquired by the optical fiber is determined. The maximum acceptance cone angle, β_max_, is given by:(3)$$\beta_{\max}=\sin^{-1}\left(\frac{\mathrm{NA}}{n_{0}}\right)$$
where n_0_ is the refractive index of the medium between the fiber end and the target surface.

The derivation of the Equations used to consider the effect of the object size on the spectral radiance is described in [33], including different cases under analysis with the integration limits as a complement to the analysis provided in [34]. Considering the transimpedance amplifier of the detector, the output voltage (V_D_), see [20], is given by:(4)$$V_{D}\left(\lambda ,T\right)=G\times I_{D}\left(\lambda ,T\right)+V_{\mathrm{noise}}$$
where V_noise_ is a factor that accounts for shot, thermal, and dark noises, as well as the offset voltage at the detector’s output, and G is the amplifier’s transimpedance gain.

If the chosen wavelength channels of a two-color optical fiber pyrometer are close enough to one another, the surface’s emissivity can be discarded. The measuring area (S_NA_) of the pyrometer, see Figure 1, depends on its acceptance angle (Ɵ_max_) and the distance (t) between the fiber and the target surface, as follows:(5)$$S_{\mathrm{NA}}={\left(t\times \tan\theta_{\max}+\frac{d_{\mathrm{core}}}{2}\right)}^{2}\times \pi$$

The spatial resolution is defined as the diameter of the light spot calculated using (5).

## 3. Experimental Set-Up

Figure 2 depicts the pyrometer set-up, which is based on that in a previous study [20]. The radiation gathered by the acceptance cone of the fiber was divided into two spectral bands with central wavelengths at around 1310 and 1550 nm by a low-insertion-loss optical filter. Two InGaAs photodetectors (PD) with transimpedance amplifiers that operate in the 800–1700 nm spectral range were used. They transform the optical radiation into an electrical signal using high gain (HG) for weak optical powers and low gain (LG) for stronger optical signals. Then, a PC-connected data acquisition card (DAQ) captured the electrical signal. The DAQ output voltage range was from zero to either 1 V or 10 V. The sampling rate was fixed at 1 kHz for both wavelength bands. The pyrometer can measure temperatures up to 600 °C using MMF and up to 1200 °C using SMF.

Two different types of standard optical fibers were employed in this work:A multi-mode optical fiber (MMF) OM1 with core/cladding diameters of 62.5/125 μm and an NA of 0.275.A single-mode optical fiber (SMF) G.652 with core/cladding diameters of 9/125 μm and an NA of 0.14.

A dry block calibrator (Jupiter 650, Isotech) with a black body (an emissivity of 0.99) was utilized for the measurements. The control unit ensures a maximum temperature stability and uncertainty of ±0.03 and ±0.17 °C, respectively. Calibration was carried out in 25 °C increments in a temperature range from 100 to 600 °C. A calibrated metallic holder was used to guarantee the position of the fiber inside the furnace and place it at a distance of 0.2 mm from the black body surface. With a sampling rate of 1 kHz and 1 kS per temperature, the DAQ was set up to measure both wavelength channels. From the recorded samples, the average output voltage and standard deviation were determined for each temperature and wavelength channel.

We employed a commercial microheater with the dimensions shown in Figure 3 to check the real spatial resolution. The gray line of the microheater were heated up, which are positive temperature coefficient resistors, by applying a voltage of 5 V. White sections (which are made of Al_2_O_3_) were then heated via heat transfer processes, and therefore, have a lower temperature compared to those of the gray lines. In order to distinguish this temperature difference, a temperature sensor should be as small as 0.25 mm.

## 4. Results and Discussion

### 4.1. Calibration

As described in the previous section, the experiments were conducted utilizing a dry block calibrator and a black body with the pyrometer in HG configuration. To adjust the mathematical formulas for the properties of the various pyrometer elements, different approximations and assumptions were taken into account during the theoretical calculations characterizing the pyrometer response, such as in [20]. Table 2 provides an overview of the factors that were taken into consideration when theoretical analysis of all the optoelectronic components for SMF was performed. We used a typical value of 0.3 dB for each connector to calculate the total connector loss. The optical filter’s insertion loss and the fiber’s attenuation were chosen as the mean value of the selected channel band. The transimpedance amplifier gain and photodetector responsiveness were both included in the conversion factor. This value depends on the conditioning circuit and was obtained in this work by conducting experimental methods, launching known optical power into the photodiodes, and measuring the output voltage with the pyrometer system. Experiments were also used to obtain the detector voltage noise. All calculations were performed with the Symbolic Math Toolbox of MATLAB.

Figure 4 shows the experimental and theoretical calibration curves for MMF and SMF configurations. The experimental results and the simulations match very well. For MMF, the voltage collected in the channel at 1550 nm is saturated for temperatures above 600 °C as it reaches the maximum voltage readable by the DAQ (10 V). This is the reason that we ensure there was a difference between the theoretical and experimental values at this point. The dynamic range of the pyrometer increases by changing the gain switch of the photodetectors from the HG condition to the LG condition. However, for this study, we used the HG configuration to take lower temperature measurements.

In order to eliminate the effect of emissivity, the two-color optical fiber pyrometer can employ the voltage ratio of the measuring channels (in our case, 1310/1550) at each calibration point. The aforementioned ratio was obtained for each temperature, subsequent to the deduction of the noise voltage from the signals in each channel. Figure 5 illustrates this ratio for both fibers in their linear range, which refers to the range in which there is no influence of DAQ saturation, and a large enough signal is present after noise reduction. The lines depicted in Figure 5 signify the linear approximation interpolated from the experimental data. These lines indicate that within the temperature range provided, the sensitivities of the pyrometer for MMF and SMF are 0.0005 and 0.0004 °C^−1^, respectively. We used this linear approximation to calculate the temperature in the following experiments. In addition, the relative temperature error ranges from 7% to 1.6% from 275 °C to 500 °C, with it being mostly below 2%. Meanwhile, for the SMF configuration, they are always below 4% in the measuring range.

### 4.2. Relationship between Distance and Spatial Resolution of the Pyrometer

In this section, we aimed to investigate the effect of the distance between the target and fiber end on the measured optical power. To achieve this, we employed the commercial microheater shown in Figure 3 and fixed it on a holder, as shown in Figure 6. The plastic jacket of the fiber was peeled and was 5 cm in size, and the fiber end was precisely cut using a high-precision fiber cleaver. The fiber end was positioned in front of the microheater at a distance of 0.05 mm from the center of a gray line of the microheater using a 3-axis positioner (Thorlabs Mitutoyo 150-801 ME). This positioner had a microcontroller with a 25 mm travel range and engraved graduations every 0.01 mm (see Figure 6). To measure the effect of the distance, we gradually increased the distance between the fiber end and the microheater with steps of 0.05 mm using the microcontroller. At each step, the optical power was measured to determine the effect of the distance on the measured power.

For the simulation, we used the model described in [33]. Considering an object size of 250 µm, we used the same pyrometer parameters as we did in the previous section, at 340 °C, and swept the distance from 0.05 to 1.2 mm with a step of 0.05 mm. Figure 7 illustrates the results of experiments and simulations for both fibers and in channel at 1550 nm.

According to the theory, the optical power is at its maximum value and is independent of distance until the diameter of the light spot (calculated from (5)) becomes larger than the diameter of the object. This is the reason for the flat part of the graphs. The optical power decreases drastically as soon as the light spot exceeds the size of the object. The reason is that a portion of the measuring area that is outside the object is does not contain emitted radiations. In our experiment, this process allowed us to collect data from the surrounding media, which is the white part of the microheater. Consequently, the total optical power is a combination of radiations from both the gray and white areas, and the resulting measured temperature is a weighted average with less accuracy that depends on the emissivities and real temperatures in these zones. In the simulations in which we only consider a single finite hot object, there is a less dramatic drop in the experiments compared to that in the simulations, which is also evidence of the lower temperatures in white regions. Nevertheless, the experimental graphs begin to show a decline in voltage at the same distance as that in the simulation results in both MMF and SMF.

### 4.3. Testing the Spatial Resolution with a Patterned-Heated Sample

In this section, we used the microheater to measure the sample’s temperature with the optical fiber pyrometer and compared the results with the temperatures measured with a commercial thermal camera (FLIR Ax35 f = 9 mm with SC kit) that has a resolution and temperature range of 320 × 256 and from −40 °C to 550 °C, respectively. The camera also has a thermal sensitivity of <0.05 °C at 30 °C and a precision of ±5 °C or ±5 % for reading. For the pyrometer tests, we positioned the fiber optic end right before the first gray line on the left side of the microheater at a distance of 0.2 mm, as shown in Figure 6. At this distance, the spot diameters of SMF and MMF are equal to 0.065 and 0.175 mm, respectively, which are both smaller than the width of the gray lines (0.25 mm). Then, we moved the fiber to the right with steps of 0.1 mm using the microcontroller and measured the temperature at each stage.

Figure 8a shows the measured voltage by the MMF pyrometer versus position. The signal shows maximums and minimums (in both channels) following the microheater pattern process and a lower voltage level at the 1310 nm channel, which is as expected. Figure 8b shows the measured signal for the SMF pyrometer. In this case, the signal at the 1550 nm channel exhibits distinct peaks and troughs, the channel at 1310 nm has also similar features but in many measuring points, and the noise distorts the signal. Some of the reliable data are in the two positions highlighted in green, as they exhibit adequate power in both channels.

The measured signals shown in Figure 8 were converted to temperature using the ratio value and compared with the calibration ratio from Figure 5. The results are shown in Figure 9a. In the case of SMF, we calculated the temperature only for two positions with the highest temperature, as in many other positions, the signal in channel 1310 nm was mostly dominated by noise and data that were not reliable. To address this issue, SMF can be utilized without a filter by using a single-color pyrometer and by gathering radiation across the entire wavelength range. However, to enable the translation of the signal to temperature in this scenario, it is crucial to have knowledge of emissivity. According to these results, the pyrometer is capable of distinguishing between high-temperature and low-temperature zones and represented a temperature pattern with the same characteristics as the microheater. The spot area for MMF is d = 0.175 mm, and the measuring step is 0.1 mm; so, the spot area is entirely within the gray lines in at least one point and within the white areas in at least two points, and the temperature can be accurately determined for these positions. However, for positions where radiation is collected from both the white and gray areas, the temperature is a weighted average. SMF measures higher temperatures at the peaks, which could be due to its better spatial resolution that ensures the measuring area is completely inside the gray line at those points and that the measured temperature is not a weighted value.

After conducting tests with the pyrometer, we positioned the thermal camera at a distance of 15 cm away from the microheater, which is the shortest distance that can provide a proper focus. At this distance, the length of each pixel is 0.435 mm, which is bigger than the width of each gray line. Nonetheless, the manufacturer indicated that a single pixel measurement may be inaccurate for various reasons (thermal camera can develop bad pixels, solar or object reflections can cause a false high reading, and distortion in optical systems can impact measurements) and recommended covering the hot area with at least 3 × 3 pixels. A measurement cursor in the software of the camera consists of 3 × 3 pixels, which results in a length of 1.3 mm for each measuring point. On the other hand, the camera software requires the input of emissivity; however, the emissivity of gray parts is unknown. We used the typical value of emissivity for Al_2_O_3_, the constructive material of the microheater in white parts, which is 0.75.

Figure 10 shows the thermal camera image of the microheater. Line 1, with a length of 5.6 mm, is depicted in the image, showing the width of the microheater in the center. Then, we moved a measurement cursor in steps of one pixel (0.435 mm) from the line’s left point to its right point. The results are shown in Figure 9b. Since the width of each measuring zone is 3 × 0.435 mm, the result shows an average temperature in all the positions, and it is not possible to see the thermal pattern such as in the case of pyrometer measurement. The reason for having lower temperatures at the ending points is that at these points, the camera captures the average between the microheater and its background, which has a much lower temperature.

If we take the two maximum points that are almost in the middle as representatives of temperature in the microheater, and not those in the background, the average temperature according to the thermal camera would be 311 °C. The average temperature of all points with accurate measurement in the microheater, according to the optical fiber pyrometer with MMF configuration, is equal to 314 °C. With the exception of having very localized measurements without the need to be aware of emissivity in the case of the pyrometer, the results are comparable.

## 5. Conclusions

We report temperature measurements with micrometer spatial resolution, independent of material emissivity, and at a 1 kHz sampling rate using two-color optical fiber pyrometers based on standard single-mode (SMF) and multi-mode optical fibers (MMF) with small core diameter and low numerical aperture in combination with associated commercially available components. For a target surface at a distance of 0.2 mm, the pyrometer has spatial resolutions of 65 and 175 µm for SMF and MMF, respectively. The experiments with a patterned microheater show that the pyrometer resolved spatial resolutions of around 100 µm and was limited by the experimental set-up. Meanwhile, a thermal camera with 320 × 256 resolution cannot resolve the 700 µm periodic temperature pattern and requires knowledge of the target’s emissivity. The two-color optical fiber pyrometer has the potential of quickly measuring temperature gradients in small areas, independent of emissivity, such as in microthermography, or in difficult-to-access areas, such as in machining processes. However, the two-color pyrometer is limited in its ability to measure temperatures below 250 °C in the MMF configuration and below around 325 °C in the SMF configuration and cannot provide a 2D map of temperature from a single measurement. We show that optical powers gathered by each channel of SMF and MMF pyrometers from hot objects of 250 µm are independent of distance until the diameter of the light spot becomes larger than the diameter of the object, at 0.9 mm and 0.4 mm, respectively, via simulations and experiments.

## Figures and Tables

**Figure 1 sensors-23-04320-f001:**
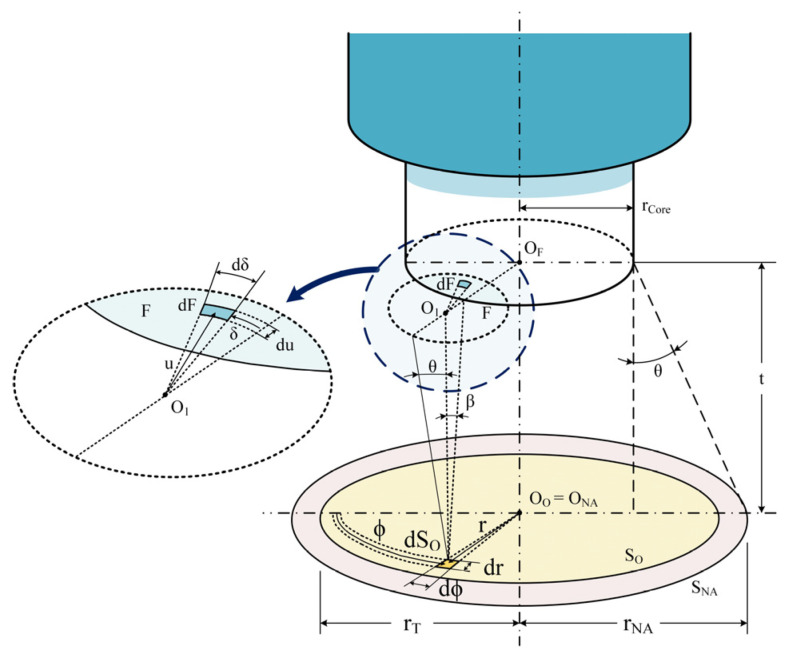
Acceptance cone of the pyrometer, target surface, and integration variables; see [27].

**Figure 2 sensors-23-04320-f002:**
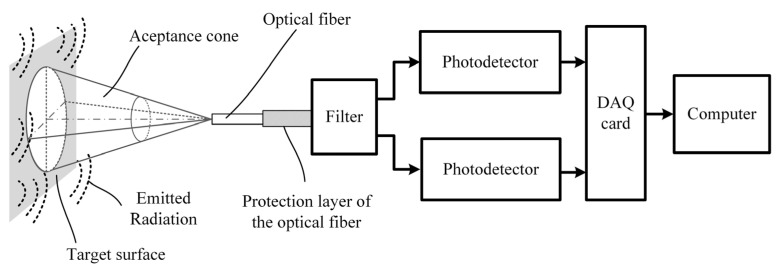
Experimental two-color optical fiber pyrometer set-up [20].

**Figure 3 sensors-23-04320-f003:**
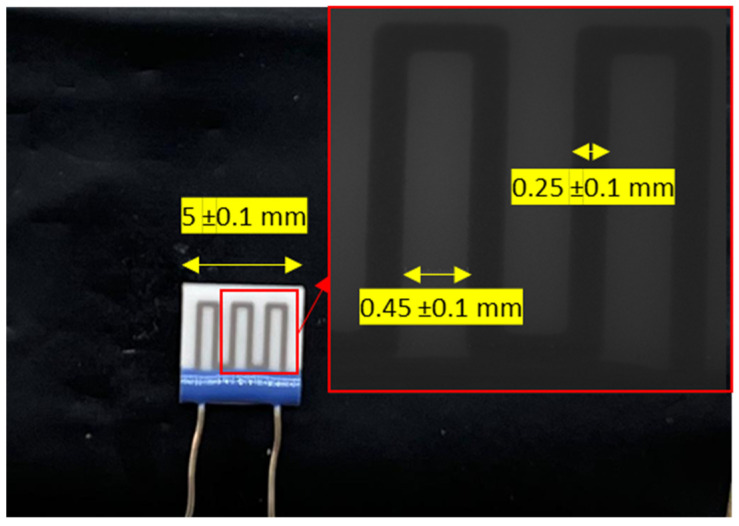
The commercial microheater and its pattern, which was acquired using an optical microscope at 100× zoom.

**Figure 4 sensors-23-04320-f004:**
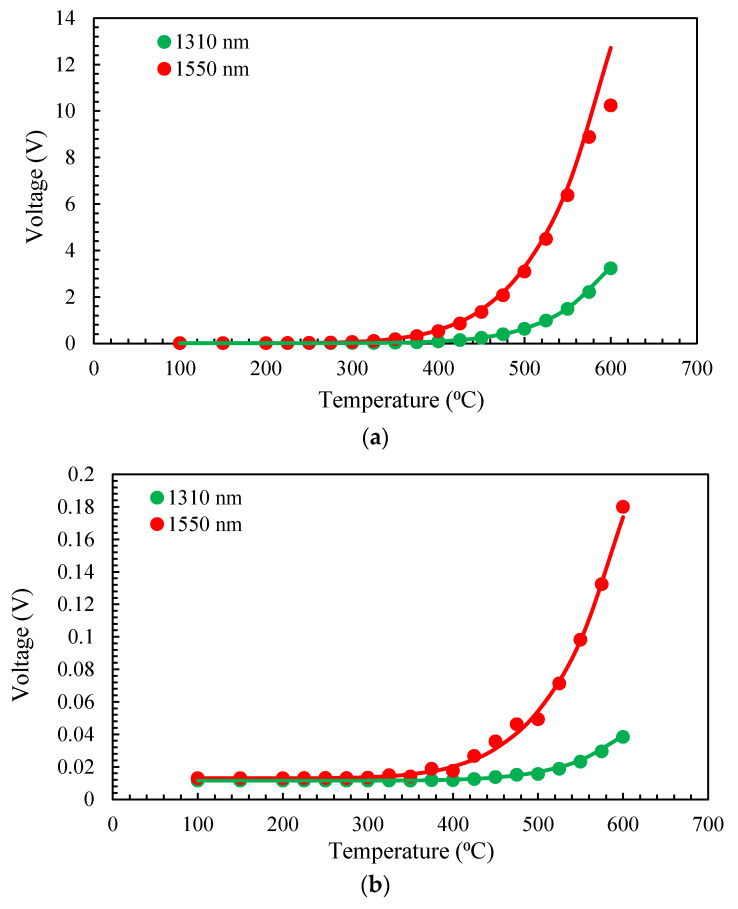
Experimental (dots) using the black body in the dry block calibrator and theoretical analysis (lines) results obtained for the pyrometer: (**a**) MMF; (**b**) SMF.

**Figure 5 sensors-23-04320-f005:**
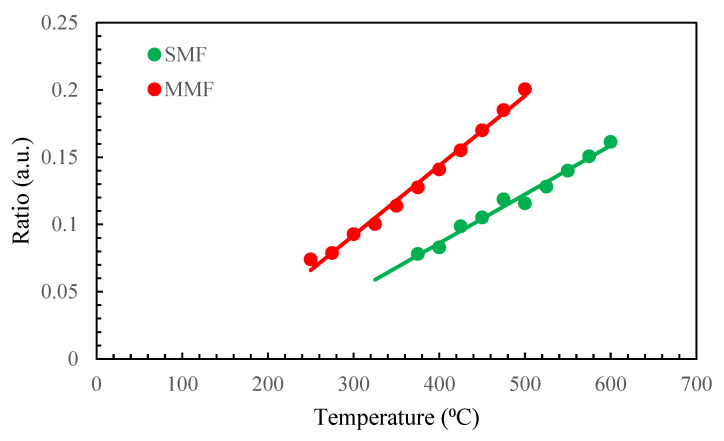
Experimental (dots) ratio and its linear approximation (lines) for SMF and MMF pyrometers.

**Figure 6 sensors-23-04320-f006:**
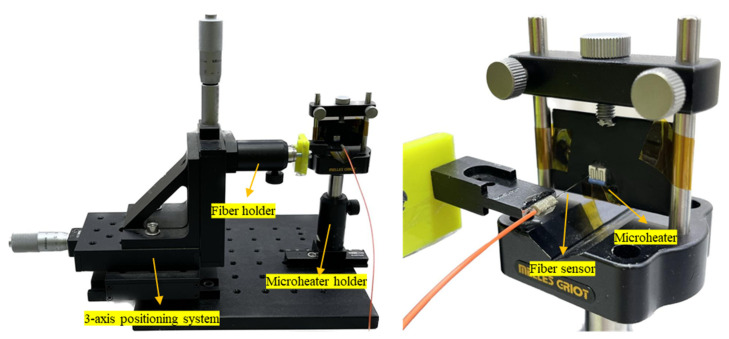
The experimental set-up with fiber and microheater holders and the positioning system.

**Figure 7 sensors-23-04320-f007:**
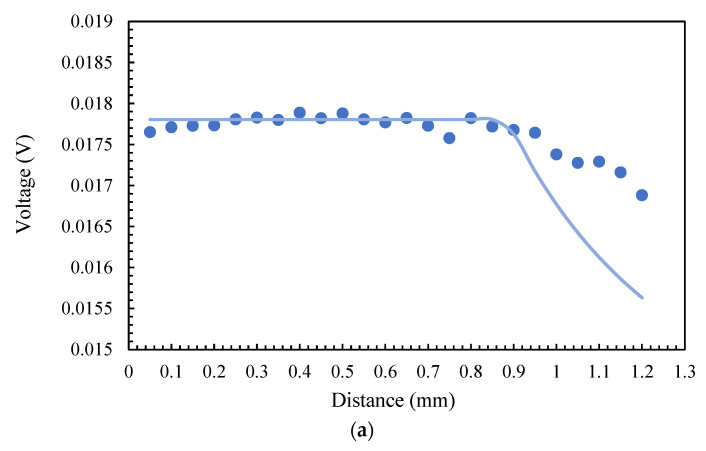
Experimental (dots) and theoretical (lines) voltage in channel 1550 nm vs. distance from an object with a diameter of 0.25 mm for (**a**) SMF, and (**b**) MMF.

**Figure 8 sensors-23-04320-f008:**
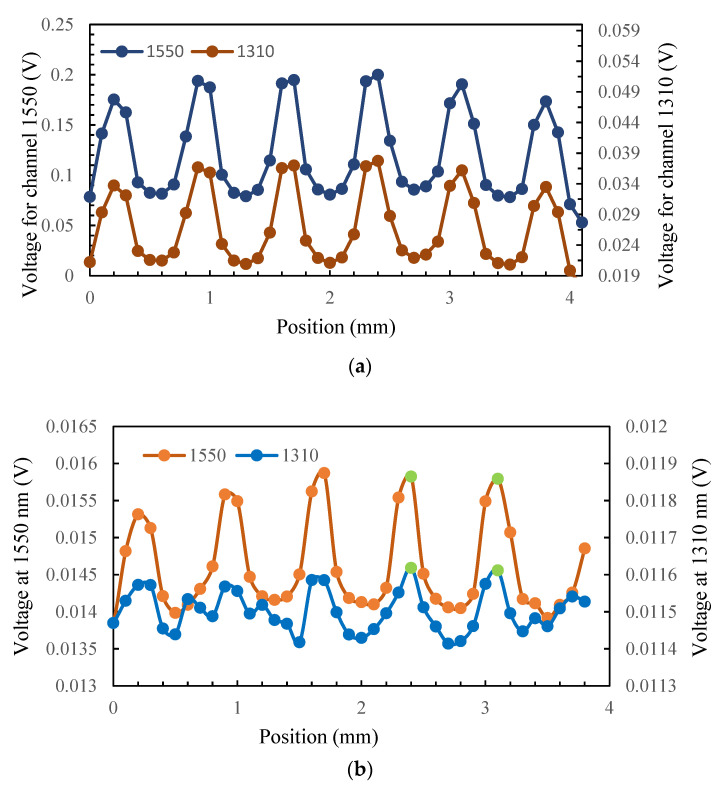
Measured voltage signal at both channels versus position conducted with pyrometer: (**a**) MMF; (**b**) SMF, green dots show data with negligible effect of noise in both channels.

**Figure 9 sensors-23-04320-f009:**
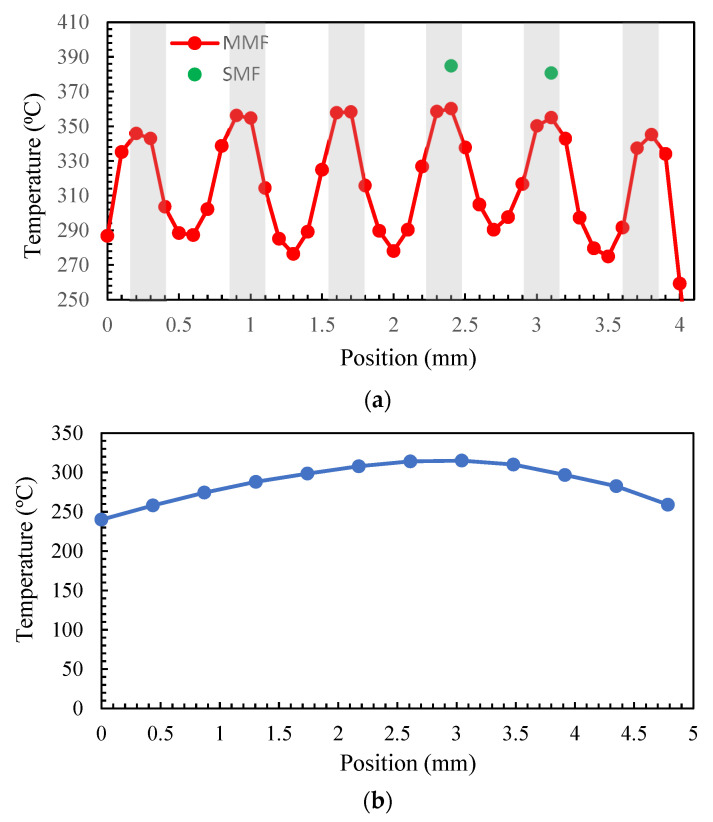
Temperature measurements with the micrometer in different positions with: (**a**) high spatial resolution optical fiber pyrometer; (**b**) thermal camera. Shadow lines show the microheater pattern.

**Figure 10 sensors-23-04320-f010:**
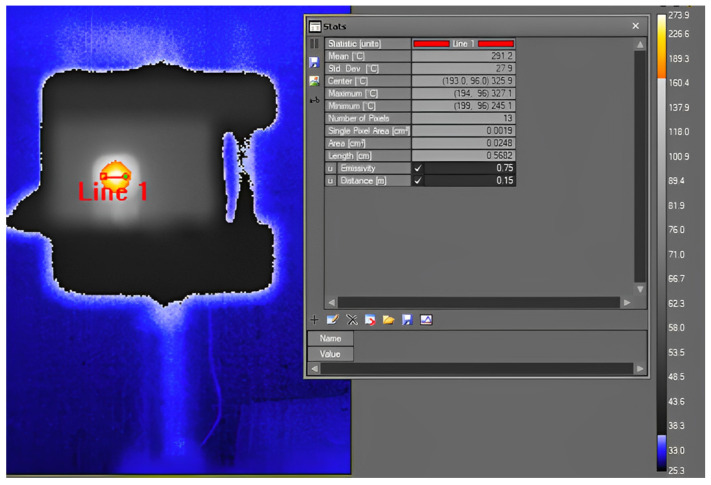
Temperature measurement of the micrometer obtained with the thermal camera.

**Table 1 sensors-23-04320-t001:** Comparison between the spatial resolution of different two-color optical fiber pyrometers found in literature and this work.

Fiber Diameter	Temperature Range	Calculated Spatial Resolution at 0.2 mm	Measured Spatial Resolution ^1^	Ref.
Silica fiber, d_core_ = 62.5 µm	2500–10,000 K	175 µm	_	[26]
F-doped fused silica fiber d_fiber_ = 330 µm	Up to 1200 °C	>330 µm	_	[27]
chalcogenide glass fiber, d_fiber_ = 400 µm	180–250 °C	>400 µm	_	[28]
Quartz fiber, d_fiber_ = 400 µm	300–1700 °C	>400 µm	_	[6]
Silica fiber, d_core_ = 9 µm	300–1200 °C	65 µm	_	[20]
Quartz optical fiber: d_core_ = 1 mm	200–1200 °C	1088 µm	_	[23]
Fluoride glass fiber d_core_ = 450 μm, d_clad_ = 500 μm	500–1000 °C	531 µm	_	[29]
Silica fiber: d_core_ = 100 μm, d_clad_ = 140 μm	900–1100 °C	188 µm	_	[30]
Pure silica core and fluorine-doped silica cladding: d_fiber_ = 400 µm		488 µm		[31]
Not mentioned	200–1200 °C	400 µm	_	[25]
Sapphire fiber, d_fiber_ = 390 µm	1000–1700 °C	>390 µm	_	[32]
silica fiber d_clad_ = 125 µm: d_core_ = 9 µm (NA = 0.14) d_core_ = 62.5 µm (NA = 0.275)	300–1200 °C	65, 175 µm	≤250 µm ^2^	This work

^1^ This parameter indicates if there is any experiment to test the spatial resolution. ^2^ Limited by the experimental set-up used (a microheater pattern with a width of 0.25 mm).

**Table 2 sensors-23-04320-t002:** Simulation parameters used for the theoretical calculations of SMF.

Parameter	Value
Total connector loss, dB	0.6
Fiber attenuation (α) at 1310 nm, dB	0.58
Fiber attenuation (α) at 1550 nm, dB	0.28
Conversion factor at 1310 nm, V/W	9 × 10^8^
Conversion factor at 1550 nm, V/W	8 × 10^8^
Voltage noise for photodetector 1, V	0.013
Voltage noise for photodetector 2, V	0.0115
Distance between target and fiber end, mm	0.2

## Data Availability

Not applicable.

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
