# Peer review of "Experimental Validation of High Spatial Resolution of Two-Color Optical Fiber Pyrometer"

_sensors, 2023, doi:10.3390/s23094320_

Round 1
Reviewer 1 Report
Review of the Manuscript Sensors-2313195
This manuscript presents the experimental results for a two-color optical fiber pyrometer with a high spatial resolution by means of temperature measurements using a patterned microheater. The work provides the experimental validation for the theoretical results presented in ref [22]:
Nunez Cascajero, A. Tapetado, and C. Vázquez, “High Spatial Resolution Optical Fiber Two Color Pyrometer with Fast Response,” IEEE Sens J, vol. 21, no. 3, pp. 2942–2950, Feb. 2021, doi: 10.1109/JSEN.2020.3022179.
Fig. 2 in ref [22] shows the spatial resolution versus the optical fiber probe distance to the target surface for three fibers with different core diameter and numerical aperture. The present manuscript demonstrates two of the different points shown in this figure. For a target distance of 0.2mm, the pyrometer has a spatial resolution of 65 and 175 μm for single-mode (9 μm - diameter core) and multimode (62.5 μm - diameter core). Theoretical results in ref [22] shows for these cases spatial resolutions of 66 and 177 μm, respectively.
The manuscript is well written and structured.
For these reasons, I recommend:
ACCEPT IN PRESENT FORM
Author Response
Manuscript ID: Sensors-2313195
Title: Experimental Validation of High Spatial Resolution in Two Color Optical Fiber Pyrometer
Dear Reviewer,
Thank you very much for your valuable comments about our manuscript.
Reviewer 2 Report
Please see the attached comments

Author Response
"Please see the attachment."

Reviewer 3 Report
The manuscript titled “Experimental Validation of High Spatial Resolution in Two Color Optical Fiber Pyrometer” by S. Safarloo et al. describes an experiment that characterises the spatial resolution of two two-colour optical fibre pyrometers. One pyrometer was constructed using a multimode fibre, while the other was constructed using a single-mode fibre. Although non-contact optical fibre pyrometry is undoubtedly an important and promising temperature sensing technique, and the manuscript is generally well-written, in my opinion, it does not contain sufficient new material to warrant publication in this journal.
Very similar versions of Figs. 1, 2, and 4 have already appeared in previous publications from the same group (see Refs. [22] and [30]). A significant portion of the information included in Table 1 overlaps with that presented in Ref. [22]. The most important new information in this manuscript is contained in Figs. 7 and 8. However, from what I can see, these two figures alone do not sufficiently justify its publication. Furthermore, the data presented in these two figures have not been analysed quantitatively or rigorously enough.
Some other issues are summarised below which include concerns about the experimental design, data analysis, and the presentation of data.
· The model, as described on pages 2 – 4 of the manuscript, is somewhat confusing. In contrast, I find the original description of the model by Ueda and co-workers almost forty years ago (T. Ueda, A. Hosokawa, and A. Yamamoto, J. Eng. Ind. 107 (1985) 127 -133) much more intuitive and revealing. I suggest that the authors carefully consider the presentation of the model and rewrite some of these paragraphs.
· The manuscript lacks some crucial information that could enhance its clarity and reproducibility. For instance, in lines 143 – 151, the authors should explicitly provide the transimpedance gains of the amplifiers, instead of using vague descriptions such as high gain and low gain. Additionally, for the models shown in Figs. 4 and 7, the authors should the provide modelling parameters. These details are essential for readers to judge and reproduce the results presented in the manuscript.
· Texts in lines 239 – 251 repeat those in lines 223 – 235.
· It is not clear which wavelength channel the data presented in Fig. 7 are based on.
· In subsection 3.1, the authors discuss how the pyrometers were calibrated, but very little information is given regarding the performance of the pyrometers. It would be helpful to include information such as the sensitivity, accuracy and the precision of the pyrometers. On the contrary, when the authors discuss the comparison between their fibre pyrometers and the FLIR thermal camera, important performance parameters of the latter are explicitly stated (see lines 256 – 257).
· I find the discussion of Fig. 8(a) in lines 264 – 272 confusing. First, it is apparent that the pyrometer designed to detect NIR radiation (1310 and 1550 nm) has higher sensitivity for samples with higher surface temperatures. It is difficult to understand why the authors did not conduct experiments at higher temperatures to reduce the impact of detector noise on their data. Second, to eliminate the effect of sample emissivity, detector noise should be subtracted from the raw data before calculating the ratio between the two detector outputs. The analysis presented in the manuscript does not explain whether this step was taken, and if not, it would be helpful if the authors could clarify the reason behind this omission.
· In lines 304 – 305, the authors claim that their two-colour fibre pyrometer has “a large dynamic range” and “fast response”. However, no data presented in the present manuscript supports such conclusion.
Author Response
"Please see the attachment."

Round 2
Reviewer 3 Report
See the attachment.

Author Response
"Please see the attachment."
